# A Versatile Punch Stroke Correction Model for Trial V-Bending of Sheet Metals Based on Data-Driven Method

**DOI:** 10.3390/ma14174790

**Published:** 2021-08-24

**Authors:** Yongsen Yu, Zhiping Guan, Mingwen Ren, Jiawang Song, Pinkui Ma, Hongjie Jia

**Affiliations:** 1Key Laboratory of Automobile Materials of Ministry of Education & School of Materials Science and Engineering, Jilin University, 5988 Renmin Street, Changchun 130022, China; yuys19@mails.jlu.edu.cn (Y.Y.); renmw@jlu.edu.cn (M.R.); songjw@jlu.edu.cn (J.S.); mapk@jlu.edu.cn (P.M.); jiahj@jlu.edu.cn (H.J.); 2International Center of Future Science, Jilin University, Changchun 130012, China

**Keywords:** V-bending, springback, punch stroke, neural network, dimensional analysis

## Abstract

During air bending of sheet metals, the correction of punch stroke for springback control is always implemented through repeated trial bending until achieving the forming accuracy of bending parts. In this study, a modelling method for correction of punch stroke is presented for guiding trial bending based on a data-driven technique. Firstly, the big data for the model are mainly generated from a large number of finite element simulations, considering many variables, e.g., material parameters, dimensions of V-dies and blanks, and processing parameters. Based on the big data, two punch stroke correction models are developed via neural network and dimensional analysis, respectively. The analytic comparison shows that the neural network model is more suitable for guiding trial bending of sheet metals than the dimensional analysis model, which has mechanical significance. The actual trial bending tests prove that the neural-network-based punch stroke correction model presents great versatility and accuracy in the guidance of trial bending, leading to a reduction in the number of trial bends and an improvement in the production efficiency of air bending.

## 1. Introduction

Sheet metal bending is a representative forming craft in manufacturing industries [1]. “Springback” refers to the elastically driven change in shape that occurs following a sheet bending when forming loads are removed from the work piece, which causes problems such as increased tolerance and variability in subsequent forming operations, in assembly, and in the final part(s) [2]. In air bending, therefore, precise bending has to be guided by a springback prediction model that represents the accurate relationship between punch stroke and forming angle [3]. However, the factors for springback of sheet metals in air bending are so complicated that the springback prediction models have a certain degree of error no matter how accurate they are [4]. In the air bending process, the forming angles need to gradually approach the expected value via repeating trial bending; that is, the punch stroke keeps being corrected until the tolerance of the forming angle is reached.

The punch stroke correction model, which affords a relationship between deviation of the bending angle and correction of punch stroke, is also critical for sheet bending and has been paid much less attention than the springback prediction models [5,6]. The deviation of the bending angle and the correction of punch stroke should be the differential or variational perturbation of the bending angle and punch stroke, respectively. Consequently, the punch stroke correction model could be regarded as the differential or variational form of the springback prediction model. If the springback prediction model presents an explicit analytical formula, the punch stroke correction model can be obtained by differentiation calculation. Some springback prediction models have been analytically deduced by means of mechanical analysis, considering the geometrical dimensions of forming dies and work pieces, the mechanical properties of sheet metals, processing parameters, etc. [7,8,9]. However, these analytical models are not accurate enough, due to simplifications and assumptions during mechanical analysis because the influence of springback in sheet bending is highly nonlinear, involving many complicated factors. Thus, their differential forms—for instance, the punch stroke correction model—are also inaccurate, leading to poor efficiency in trial bending.

With the development of machine learning, data-driven statistical models have been proposed for the prediction of springback in sheet metal bending [10,11,12,13,14]. The accuracy of the data-driven models depends on the large scale of data. Based on the design of the experiments, the data acquisition is generally implemented through a large number of real/virtual tests of air bending, considering the variations in each factor related to springback. Compared with real tests, the virtual bending tests by finite element modeling are suitable for the data-driven statistical models, due to their higher efficiency and lower costs [15]. In practical application, the data-driven statistical models present high accuracy in the prediction of springback. The accurate springback prediction models were built by approximation methods, such as response surface methodology [16,17], artificial neural network [18,19,20,21,22], and Kriging [23]. The authors proposed a springback prediction model for air bending through the combination of genetic algorithm and backpropagation neural network (GA–BPNN), presenting high accuracy and great versatility. 

However, it is infeasible to deduce the punch stroke correction model through differentiation calculation from the springback prediction models, which have remarkably complicated statistical forms. In the study, therefore, we aimed to propose a punch stroke correction model for trial tests in metal sheet bending, which would be able to guide the accurate correction of punch stroke with high efficiency. The model would be built from big data on springback in air bending through a GA–BPNN approach. Firstly, a large number of virtual bends would be implemented to obtain the big data. For comparison, additionally, another punch stroke correction model would be proposed based on dimensional analysis; that is, a semi-analytical method [24]. Finally, practical sheet metal trial bending tests would be implemented in order to investigate and contrast the correction accuracy of punch stroke in the two models.

## 2. Research Methods

### 2.1. Modeling Principle

In order to establish a large-scale dataset, the virtual bending finite element tests were simulated. On the basis of the acquired dataset, the GA–BPNN prediction model was trained. The input parameters consisted of bending angle $\alpha^{0}$, elastic modulus *E*, yield strength $\sigma_{s}$, hardening coefficient *K*, hardening exponent *n*, thickness *t*, and groove width $B_{V}$. The output parameter was punch stroke *D*.

Random angle deviations $\Delta \alpha^{0}$ were generated and used to alter the original bend angles $\alpha^{0}$ in the established dataset. Corresponding with new angles, the new punch strokes were obtained through calculation of the prediction model. Punch stroke compensation ΔD was defined as the offset before and after alteration. In this way, another large sample dataset was created that contained angle deviations $\Delta \alpha^{0}$ and punch stroke compensations ΔD. According to the new dataset, a GA–BPNN and dimensional analysis were used to build the correction model.

#### 2.1.1. GA–BPNN Model

Machine learning is a statistical modeling technique that enables a computer system to learn or recognize implicit relationships in given data without any explicit description. Abstract information or undiscovered phenomena can theoretically be modeled by means of machine learning as a result of its data-driven properties [25,26,27].

Among diverse machine learning algorithms, artificial neural networks (ANNs) are popular because of their excellent modeling performance and wide range of applications for general approximation [28,29,30]. The backpropagation neural network (BPNN) is a sort of ANN that is widely applied at present; its basic idea is to adjust and modify the connection weights and thresholds of the network through the reverse propagation of network output errors, so as to minimize the mean squared error of the output. 

In general, the initial weight and bias of the network are generated randomly at the beginning of network training, which may result in local optima of the network. In this paper, a genetic algorithm (GA) was used for optimization the initial weight and bias of the network [31]. Based on a GA–BPNN, the prediction model and correction model were obtained in this paper. The network model is shown in Figure 1.

(1)Forward propagation of signals

The input variables need to be normalized to avoid adverse factors in the optimization process, which is denoted as $x=a^{\left(0\right)}$. The feedforward neural network propagates information by iterating the following formulae:(1)$$z^{\left(l\right)}=w^{\left(l\right)}a^{\left(l-1\right)}+b^{\left(l\right)}$$
(2)$$a^{\left(l\right)}=f_{l}\left(z^{\left(l\right)}\right)$$
where $z^{\left(l\right)}$ is the net input of neurons in layer l; $w^{\left(l\right)}$ is the weight matrix from layer l−1 to layer l; $a^{\left(l\right)}$ is the output of neurons in layer l; $b^{\left(l\right)}$ is the bias from layer l−1 to layer l; and $f_{l}\left(.\right)$ is the activation function of neurons in layer *l*. Equations (1) and (2) can also be combined and written as:(3)$$a^{\left(l\right)}=f_{l}\left(w^{\left(l\right)}a^{\left(l-1\right)}+b^{\left(l\right)}\right)$$

The final output of the network $a^{\left(L\right)}$ can be obtained through the layer-by-layer transmission of information in the feedforward neural network. The whole network can be regarded as a compound function φ(x; w, b). The input at the first level is defined as  x, and the output of the whole function is $a^{\left(L\right)}$.
(4)$$x=a^{\left(0\right)}\to z^{\left(1\right)}\to a^{\left(1\right)}\to \cdots \to z^{\left(L\right)}\to a^{\left(L\right)}$$
where w and b represent the weights and bias, respectively, of all layers in the network.

(2)Backpropagation

Each sample $x^{\left(n\right)}$ in the given training set $D=\left\{\left(x^{\left(n\right)},y^{\left(n\right)}\right)\right\}\begin{array}{c} N\\ n=1 \end{array}$ is input to the feedforward neural network, and then ${\hat{y}}^{\left(n\right)}$ can be obtained, whose structural risk function of the dataset D is defined as:(5)$$Rw,b=\frac{1}{N}\sum_{n=1}^{N}\varsigma \left(y^{\left(n\right)},{\hat{y}}^{\left(n\right)}\right)+\frac{1}{2}\lambda ∥w∥\begin{array}{c} 2\\ F \end{array}$$
where w and b are all the weight matrices and bias vectors in the network respectively; $∥w∥\begin{array}{c} 2\\ F \end{array}$ is the regularization term to prevent overfitting; λ>0 is the super parameter, and the larger λ is, the closer w is to 0.

The parameters of network can be learned through the gradient descent algorithm. In each iteration of the gradient descent algorithm, the parameters $w^{\left(l\right)}$ and $b^{\left(l\right)}$ of the l layer are updated as follows:(6)$$w^{\left(l\right)}\leftarrow w^{\left(l\right)}-\eta \frac{\partial Rw,b}{\partial w^{\left(l\right)}}=w^{\left(l\right)}-\eta \left(\frac{1}{N}\sum_{n=1}^{N}\frac{\partial \varsigma y^{\left(n\right)},{\hat{y}}^{\left(n\right)}}{\partial w^{\left(l\right)}}+\lambda w^{\left(l\right)}\right)$$
(7)$$b^{\left(l\right)}\leftarrow b^{\left(l\right)}-\eta \frac{\partial Rw,b}{\partial b^{\left(l\right)}}=b^{\left(l\right)}-\eta \left(\frac{1}{N}\overset{N}{\underset{n=1}{\sum }}\frac{\partial \varsigma y^{\left(n\right)},{\hat{y}}^{\left(n\right)}}{\partial b^{\left(l\right)}}\right)$$
where η is the learning rate.

#### 2.1.2. Mathematical Model of Dimensional Analysis

Dimensional analysis is an analytical method to establish mathematical models in the field of physics [32,33]. Based on dimensional analysis, the laws of physics can be explained by comparing the dimensions of independent and dependent variables. According to the principle of homogeneity of dimensions, the dimensions on both sides of the equals sign must be the same when mathematical expressions are used to express physical relations. The Buckingham π theorem can be expressed in a physical equation with n variables as: (8)$$f(x_{1},x_{2},x_{3},\ldots x_{n})=0$$
where m variables are independent of one another, and the remaining (n − m) variables are independent. The physical relations can be expressed by (n − m) dimensionless variables as follows:(9)$$F(\pi_{1},\pi_{2},\pi_{3},\ldots \pi_{n-m})=0$$
where $\pi_{1},\pi_{2},\pi_{3},\ldots \pi_{n-m}$ are (n − m) dimensionless variables. The main factors that affect the physical process should be ascertained. According to the dimensionless method, the functional relationship between the factors can be established. Through the combination of experiments and functional relationships, the exact mathematical expression is obtained.

### 2.2. Sample Range Definition for Dataset

The springback of sheet metal is affected by many factors, including material parameters, dimensions of V-dies and blanks, and processing parameters. To simplify the modeling process, in this paper, it is assumed that the sheet metal materials are independent of strain rate and strain path, obeying the Hill’48 anisotropic yield criterion and the Hollomon hardening model. The functional form of the hardening model can be expressed as follows:(10)$$\{\begin{array}{c} \sigma =E\varepsilon \;\;\;\;\;\;(\varepsilon <\varepsilon_{0})\\ \sigma =K\varepsilon^{n}\;\;\;\;(\varepsilon >\varepsilon_{0}) \end{array}$$
where *E* is the elastic modulus, *K* is the hardening coefficient, and *n* is the hardening exponent. Three above and the yield strength $\sigma_{s}$ were regarded as material factors. For a common bending process, an 88° V-die is usually desirable. The processing parameters include width of slot $B_{V}$ and punch stroke *D*, and the product factor is the thickness of sheet metal T. As mentioned above, 7 affecting factors were involved. 

According to the distribution of sheet metal properties, the conventional working conditions of the bending process, and the standard thicknesses of sheet metals, the variation ranges of the 7 factors were determined under various conditions, as shown in Table 1.

A Latin hypercube design was used to determine the sample distribution of the 7 factors. A total of 1732 combinations were obtained for the springback prediction model. It is important to note that the sample size of this article is only an example of the large sample number, and the actual sample can choose a larger number.

### 2.3. Acquisition of Finite Element Sample for Training Data

As the output of the prediction model, the springback angle of each combination was obtained via the finite element simulations of sheet metal bending. In order to simulate V-bending and springback, a combination of explicit and implicit methods was used. The element type of the sheet metal part was four-node shell element (S4R), and the friction coefficient was 0.1.

Perpendicular to the direction of the bending line, the minimum mesh size was 0.2 mm and the maximum size was 2.0 mm. Parallel to the direction of the bending line, the mesh size was 0.6 mm. Five integration points were set in the direction of thickness. The simulation process of bending and springback is shown in Figure 2. Each angle after springback corresponding to each combination of factors was obtained. A total of 20 samples were randomly selected as test samples; samples for modeling and for testing are shown in Table 2 and Table 3, respectively.

All calculations were performed on a personal computer (LATOP-FM0CIDDQ Intel(R) Core(TM) i7-108575H CPU @ 2.30GHz(16CPUs), ~2.3GHz). Dataset was obtained through integration of ABAQUS and Isight. The GA-BPNN was established using MATLAB 2016a.

### 2.4. Mechanical Tests and Bending Tests

Five different sheet metals were selected for the tests, including mild steel HC220YD, stainless steel 304, aluminum alloy 5182, high-strength steel DP980, and copper H62. The widths of samples were processed to 20 mm. The sheets mentioned above were used for uniaxial tensile tests and bending experiments. The uniaxial tensile tests were performed on the electronic universal material testing machine (AGS-100kN, Shimadzu, Suzhou, China). The material performance parameters were obtained as shown in Table 4.

Sheet metal air bending experiments were carried out with a computerized numerical control bending machine (WDB100-3100, JFMMRI-JIEMAI, Jinan, China). The V-shaped slot angle was 88°, the punch radius r was 1 mm, the V-shaped slot width $B_{V}$ was 12 mm, and the punch round radius r was 1 mm. The punch stroke was measured with a grating ruler whose accuracy was within 2 µm. Bending molds and parts after forming are shown in Figure 3.

Each sheet metal was subjected to four bending tests with different punch strokes. After bending, the springback angles were measured by angle ruler with ±0.08° accuracy, the data from which are listed in Table 5. The accuracy of the simulations was verified by the comparison with the actual bending tests.

### 2.5. Data Acquisition for the Correction Model

The remaining samples were divided into 85% and 15% as training samples and verification samples, respectively. Then, the GA-BPNN springback prediction model with 7 factors was established. *E*, $\sigma_{s}$, *K*, *n*, *t*,$\;B_{V},$ and $\alpha_{0}$ were input parameters, and punch stroke D was the output parameter. The mean squared error (MSE) was used to measure the accuracy of the network:(11)$$MSE=\frac{1}{n}\overset{n}{\underset{i=1}{\sum }}{\left(y_{i}-{\hat{y}}_{i}\right)}^{2}$$

The conclusion of the network structure research showed that the error was minimal in the network with the [1,2,3,4,5,6,7] structure. The comparisons between the simulation samples (Table 3) and the network prediction results, as well as the bending tests (Table 5) and the network prediction results, are shown in Table 6. As can be seen from Table 6, the prediction deviation of the punch stroke prediction model was within 0.16 mm.

In general, to achieve a target bending angle, a trial bending needs to be carried out. Then, according to the difference from the target value, the forming angle is adjusted by the correction of the punch stroke and, therefore, the difference from the target value can be reduced. It usually takes three or four attempts to reach the target angle. Our punch stroke prediction model could ensure that the error of stroke prediction was within 0.2 mm. It was our aim to control the final sheet’s forming angle by fine-tuning the punch stroke, which is also the significance of the correction model. Based on the prediction model, the punch strokes $D^{1}$ were obtained. A total of 1732 data points as $\Delta \alpha^{0}$ were randomly generated and evenly distributed in [−3°, 3°]. Each $\alpha_{0}$ was changed to $\alpha_{0}+\Delta \alpha^{0}$ correspondingly. New angles were input to the prediction model, and the new punch strokes $D^{2}$ were generated. Stroke difference ΔD was defined as:(12)$$\Delta D=D^{2}-D^{1}$$

According to $\Delta \alpha^{0},$ with the other corresponding factors as input and ΔD as output, the correction model could be established. Some training samples are shown in Table 7, and testing samples are shown in Table 8.

## 3. Results and Analysis

### 3.1. Punch Stroke Correction Model Based on a GA-BPNN

To optimize the topology of the neural network, different hidden layers and different neurons in hidden layers were studied (random weights and bias were used tentatively). First, we focused on the springback prediction model. The MSE values (sum of the mean squared errors of the training set and verification set) under different structures were obtained, as shown in Figure 4. Early stopping was used to ensure that the model was not overfitting.

Compared with the nets containing one or two hidden layers, the accuracy of the network with three hidden layers is higher, and is also more stable. The MSE of the network with a [1,2,3,4,5,6,7] architecture can be less than 1.258 (°^2). According to the parameters of the network structure study, we determined a [1,2,3,4,5,6,7] fully connected architecture as the BPNN structure of the prediction model. The total number of network parameters thus determined was 545 (128 + 128 + 272 + 17). Then, a genetic algorithm was used to optimize the initial weights and bias of the network.

Based on experience from other studies and trial training, the following parameters worked well. A summary of the GA parameters is shown in Table 9.

The same strategy and parameters were adopted for the punch stroke correction model, which is also a network with seven inputs and a single output. To illustrate the advantage of the GA in the stability of optimization, the decreasing loss trend of the training and validation datasets in the GA-BPNN correction model is shown in Figure 5.

The GA-BPNN combines the advantages of efficiency and accuracy. After initial parameter optimization, only 5838 epochs are trained to reach the target, which is just half of the training process before GA optimization (12,483). After network training, the MSE could be less than 2.6872 × 10^−4^ mm^2^.

The regression coefficient of network training is shown in Figure 6. Comparison between testing samples and network-predicted values is shown in Figure 7a, and deviation of punch stroke compensations is shown in Figure 7b.

As shown in Figure 6 and Figure 7, the deviation of the punch strokes can be controlled within 0.05 mm. From the above results, it can be concluded that the neural network model can correct the punch stroke with sufficient accuracy.

### 3.2. Punch Stroke Correction Model Based on Dimensional Analysis

In this work, the functional relationship between punch stroke compensation ΔD and angle deviation $\Delta \alpha^{0}$ with elastic modulus *E*, yield strength $\sigma_{s}$, hardening coefficient *K*, hardening exponent *n*, sheet thickness *t*, and groove width $B_{V}$ could be expressed as: (13)$$f(\Delta \alpha^{0},\Delta D,B_{V},t,K,E,\sigma_{s},n)=0$$

The basic dimensions—including length (L), mass (M), and time (T)—were used. $\Delta \alpha^{0}$ and n are dimensionless, while the other physical quantity can be expressed as:$$\left[\Delta D\right]=L,\;[B_{V}]=L,\;[t]=L,\;\left[K\right]=L^{-1}{\mathrm{MT}}^{-2},$$
$$\left[E\right]=L^{-1}{\mathrm{MT}}^{-2},\;\left[\sigma_{s}\right]=L^{-1}{\mathrm{MT}}^{-2}$$

According to the π theorem, four dimensionless variables could be obtained from four fundamental solutions:$$\pi_{1}=\Delta D^{1}B_{V}{}^{0}t^{-1}K^{0}E^{0}\sigma_{s}{}^{0}=\frac{\Delta D}{t}$$
$$\pi_{2}=\Delta D^{0}B_{V}{}^{1}t^{-1}K^{0}E^{0}\sigma_{s}{}^{0}=\frac{B_{V}}{t}$$
$$\pi_{3}=\Delta D^{0}B_{V}{}^{0}t^{0}K^{1}E^{-1}\sigma_{s}{}^{0}=\frac{K}{E}$$
$$\pi_{4}=\Delta D^{0}B_{V}{}^{0}t^{0}K^{0}E^{-1}\sigma_{s}{}^{1}=\frac{\sigma_{0.2}}{E}$$

And $\Delta \alpha^{0}$ and n were written as:$$\pi_{5}=n$$
$$\pi_{6}=\Delta \alpha^{0}$$

Therefore, there was a function φ as:$$\frac{\Delta D}{t}=\varphi \left(\frac{B_{V}}{t},\frac{K}{E},\frac{\sigma_{s}}{E},n,\Delta \alpha^{0}\right)$$
whose specific expression form was:(14)$$\Delta D=a_{0}t{\left(\frac{B_{V}}{t}\right)}^{a_{1}}{\left(\frac{K}{E}\right)}^{a_{2}}{\left(\frac{\sigma_{s}}{E}\right)}^{a_{3}}{\left(n\right)}^{a_{4}}{\left(\Delta \alpha^{0}\right)}^{a_{5}}$$

In order to avoid errors caused by different orders of magnitude between factors, each dimensionless variable was normalized before calculation. The least squares method was used for fitting. The parameters obtained by fitting were $a_{0}$ = 0.41426, $a_{1}\;$ = 0.01508, $a_{2}$ = 0.0285, $a_{3}$ = 0.0302, $a_{4}$ = 0.00994, and $a_{5}$ = −0.1584. The specific function was obtained as follows:(15)$$\Delta D=0.41426t{\frac{B_{V}}{t}}^{0.01508}{\frac{K}{E}}^{0.0285}{\frac{\sigma_{s}}{E}}^{0.0302}{\left(n\right)}^{0.00994}{\left(\Delta \alpha^{0}\right)}^{-0.1584}$$

The model was tested with the data shown in Table 8. Comparison between the test samples and the predicted values of the model is shown in Figure 8a, and the deviation of the punch stroke compensation value is shown in Figure 8b.

Comparing the results, using the punch stroke correction model based on dimensional analysis, the deviation of punch stroke compensation can be kept within 0.15 mm, while the punch stroke correction model based on a GA-BPNN can keep it within 0.05 mm. The GA-BPNN model can predict punch stroke more accurately and control the forming angle to be closer to the target angle.

### 3.3. Application Examples

Three kinds of sheet metal were chosen for bending experiments with the universal testing machine (WQ4200, Changchun Kexin instrument institute, Changchun, China)—HC220YD mild steel, 304 stainless steel, and 5182 aluminum alloy—to further illustrate that the GA-BPNN punch stroke correction model could control the forming angle accurately by adjusting the punch stroke. The mechanical property parameters of the three materials are shown in Table 4.

Three target angles were chosen for each material. The initial strokes were obtained according to the target angle by calculation of the GA-BPNN punch stroke prediction model. Based on the GA-BPNN punch stroke correction model, the strokes were adjusted by the deviation of the angles. The bending tests were performed with the universal material testing machine, as shown in Figure 9, so that the punch stroke could be freely controlled. The punch strokes were measured on a grating scale. For the measurement of the forming angles, a digital protractor was used. The radius of punch *R* was 1 mm; the width of the V-shaped groove $B_{V}$ was 12 mm, and the radius of the punch fillet R was 1 mm. 

The bending samples are shown in Figure 10, and the experimental data are shown in Table 10. Some samples could reach the target angles directly through the prediction model, while most samples could reach the target angles through one use of the correction model. All samples from the tests could achieve error precision within 0.5°.

It should be noted that the main purpose of this paper is to provide a correction method for studying the sheet metal bending springback, which cannot represent optimal accuracy. In practice, the accuracy of the correction model can be improved by further improving the finite element simulation accuracy and machine learning fitting accuracy to meet the requirements of actual conditions.

## 4. Conclusions

To improve the forming accuracy of air V-bending, this paper establishes a punch stroke correction model by means of a GA-BPNN and dimensional analysis. The correction results of the semi-analytical model and the machine learning model are compared based on the actual bending test. The correction model with guaranteed accuracy can provide a more accurate machining stroke for actual production to minimize the shape defects caused by springback. The following conclusions can be drawn from this study:(1)A large sample dataset was established via finite element method for bending experiments using various sheet metals. Based on the dataset, a GA-BPNN prediction model was established, whose accuracy was guaranteed within 0.16 mm by the contrast with actual bending experiments;(2)In order to further improve the accuracy of the model-guided processing, the GA-BPNN and dimensional analysis were used for the establishment of the correction model. By comparing the verification results with the targets in the dataset, the GA-BPNN correction model was more capable of fitting the target problem, and the deviation of punch stroke compensation could be controlled within 0.05 mm;(3)The accuracy of the GA-BPNN punch stroke prediction model and the GA-BPNN punch stroke correction model was verified via bending experiments using a universal material testing machine. Calculated by the prediction model once and the correction model two times, the error of all of the forming angles could be less than 0.5°. Our work provides a new method to solve the problem of precise rapid forming in sheet metal bending.

## Figures and Tables

**Figure 1 materials-14-04790-f001:**
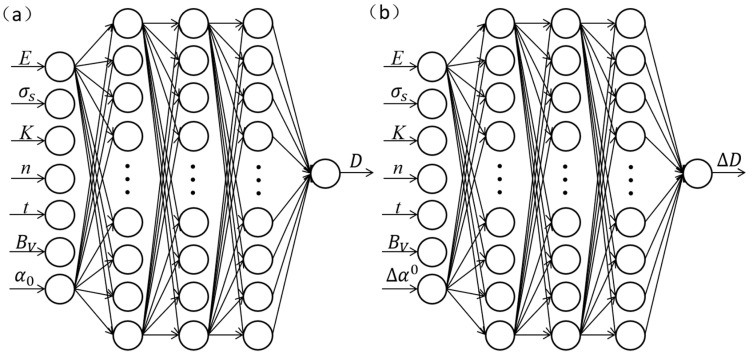
Structural diagram of BPNN: (**a**) punch stroke prediction model and (**b**) bending correction model.

**Figure 2 materials-14-04790-f002:**
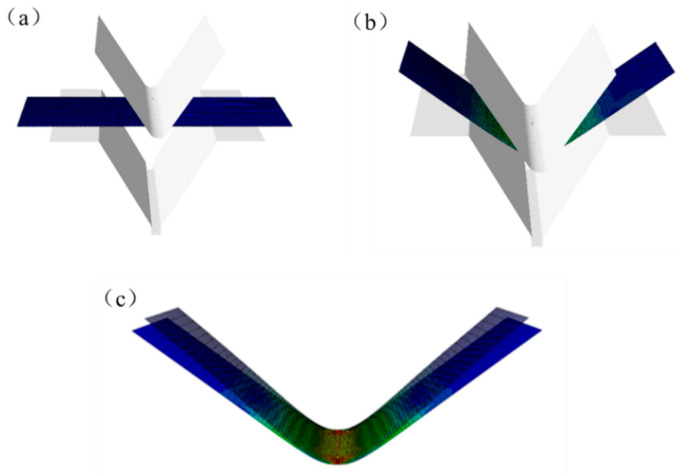
Finite element simulation process (**a**) before bending, (**b**) after bending, and (**c**) after springback.

**Figure 3 materials-14-04790-f003:**
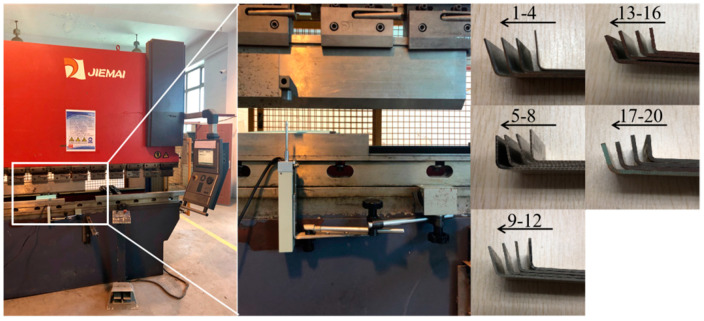
Bending machine and bending parts.

**Figure 4 materials-14-04790-f004:**
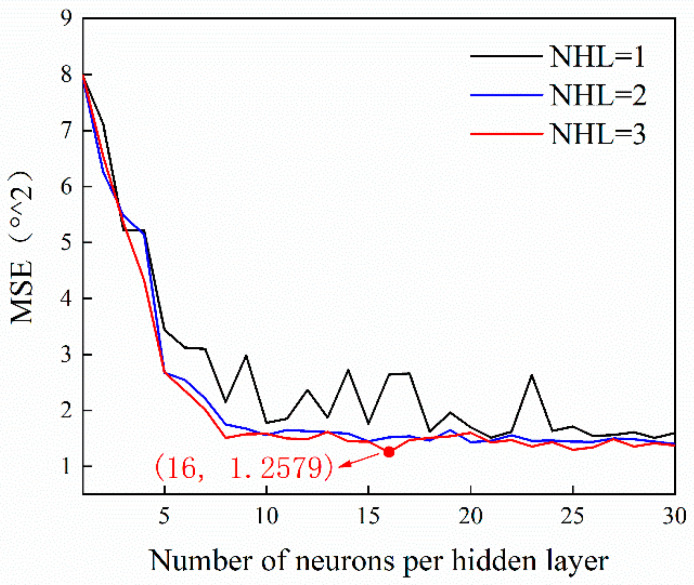
Results for MSE after 10 training runs as a function of hidden layers and neurons per layer.

**Figure 5 materials-14-04790-f005:**
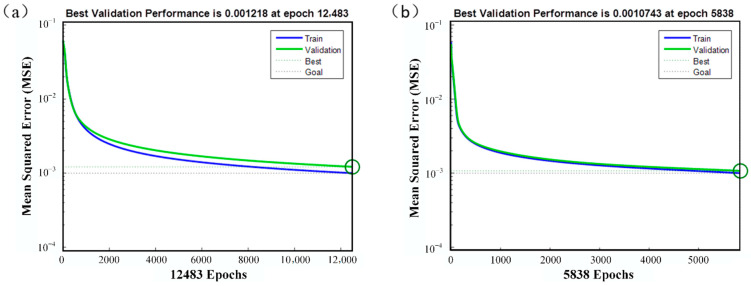
MSE downtrend comparison before and after GA optimization: (**a**) before optimization; (**b**) after optimization.

**Figure 6 materials-14-04790-f006:**
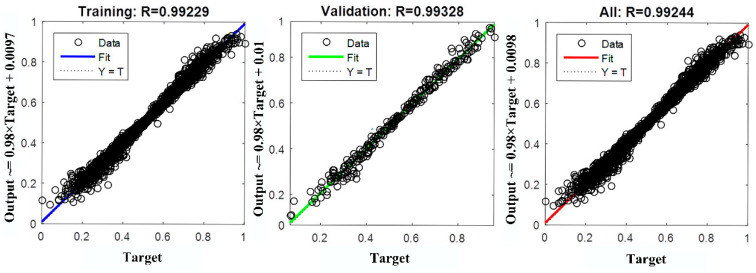
Regression values of network training.

**Figure 7 materials-14-04790-f007:**
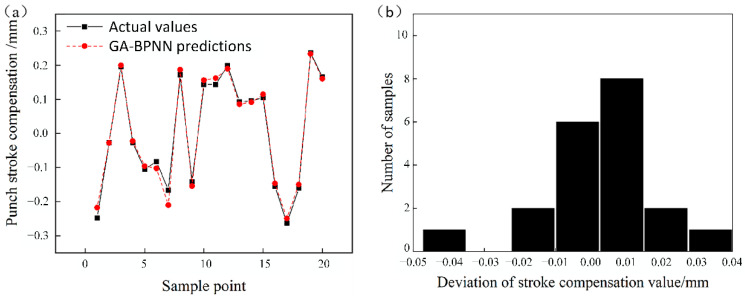
(**a**) Comparison between testing samples and the predicted compensations of the correction model. (**b**) Stroking deviation.

**Figure 8 materials-14-04790-f008:**
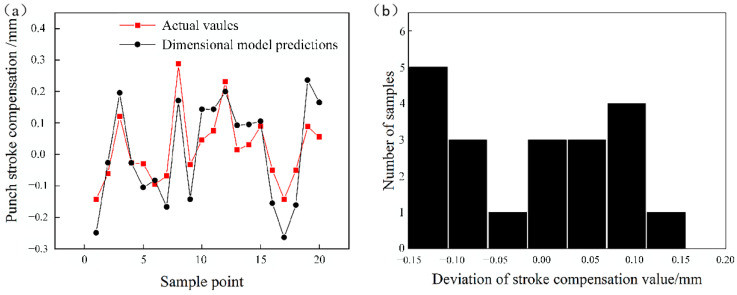
(**a**) Comparison of the testing samples with the predictions of the dimensional model, and (**b**) deviation of the punch stroke compensation.

**Figure 9 materials-14-04790-f009:**
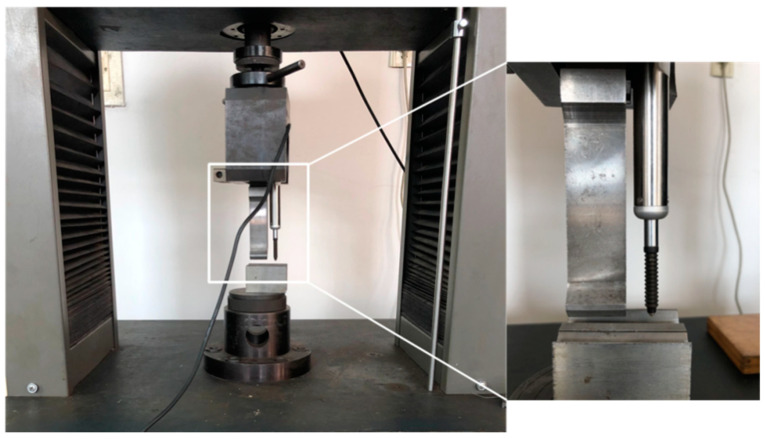
Bending experiment with the universal testing machine.

**Figure 10 materials-14-04790-f010:**
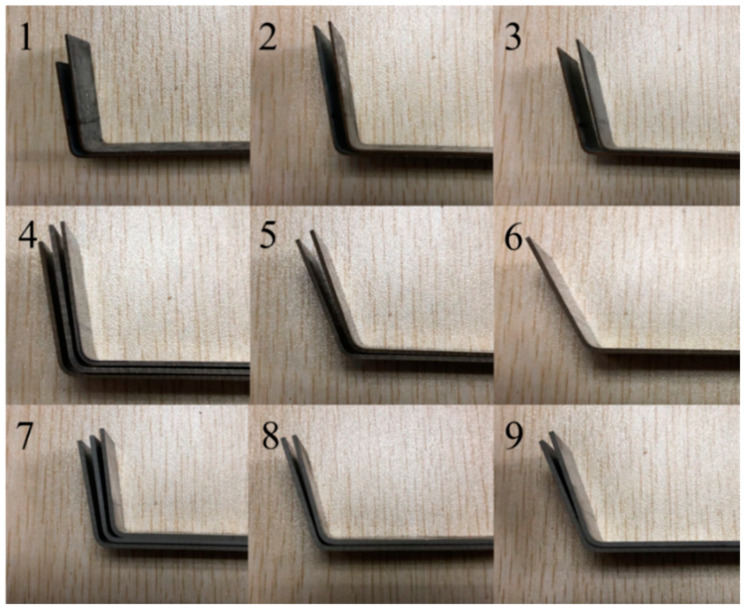
Bending samples, where 1–3 are HC220YD steel, 4–6 are 304 stainless steel, and 7–9 are 5182 aluminum.

**Table 1 materials-14-04790-t001:** The range of the factors.

	*D/*mm	*t/*mm	*E/*GPa	$B_{V}/\mathrm{mm}$	*K/*MPa	*n*	$\sigma_{s}/\mathrm{MPa}$
1	3–5	0.6–2	70–220	12	500–2000	0.1–0.6	120–1000
2	3–5.5	0.6–2	70–220	14	500–2000	0.1–0.6	120–1000
3	3.5–6.5	0.6–2	70–220	16	500–2000	0.1–0.6	120–1000
4	4–7	0.6–3	70–220	18	500–2000	0.1–0.6	120–1000
5	5–8	0.6–3	70–220	20	500–2000	0.1–0.6	120–1000

**Table 2 materials-14-04790-t002:** Modeling samples from the simulation dataset.

Number	*D*/mm	*E/*GPa	$B_{V}/\mathrm{mm}$	*K/*MPa	*n*	*t/*mm	$\sigma_{s}/\mathrm{MPa}$	$\alpha_{0}$
1	7.89	75.263	20	695.49	0.369	1.43	510.38	101.05
2	5.94	116.992	20	1507.52	0.136	2.72	843.41	112.83
3	5.67	77.142	16	1218.05	0.297	1.63	126.62	93.76
4	5.50	171.503	18	1342.11	0.148	1.95	344.96	106.17
5	3.68	115.112	16	1390.98	0.425	0.73	263.36	126.52
6	7.48	97.067	20	921.05	0.542	1.88	772.83	101.03
7	3.74	104.210	14	1563.91	0.516	1.85	891.93	118.49
8	3.99	166.992	12	1278.20	0.110	1.17	437.59	101.99
9	5.69	116.616	16	1078.95	0.212	1.85	534.64	98.77
10	3.52	153.834	12	1943.61	0.309	1.72	982.36	110.29
11	5.61	168.496	18	1887.22	0.182	2.00	618.45	106.71
12	7.04	99.323	20	1526.32	0.342	1.44	201.60	96.49
13	6.98	122.631	20	1424.81	0.225	2.80	953.68	102.52
14	5.53	143.684	20	1578.95	0.149	2.84	889.72	117.32
15	4.76	194.812	14	1913.53	0.248	1.37	631.68	102.06
16	6.71	111.353	20	1011.28	0.108	2.69	192.78	101.53
17	3.71	196.315	14	1921.05	0.267	1.08	146.47	110.28
18	6.91	191.052	20	902.26	0.103	1.02	375.84	104.22
19	4.19	155.338	12	1116.54	0.491	0.87	847.82	101.42
…	…	…	…	…	…	…	…	…
1712	4.94	189.172	18	1443.61	0.128	1.50	532.43	116.43

**Table 3 materials-14-04790-t003:** Testing samples from the simulation dataset.

Number	*D*/mm	*E/*GPa	$B_{V}/\mathrm{mm}$	*K/*MPa	*n*	*t/*mm	$\sigma_{s}/\mathrm{MPa}$	$\alpha_{0}$
1	6.02	114.736	18	1778.2	0.425	1.75	589.77	103.63
2	3.97	123.007	12	1330.83	0.432	1.15	554.49	99.42
3	4.68	107.969	16	1872.18	0.524	1.01	298.65	117.55
4	3.26	103.834	12	511.28	0.504	0.78	137.64	166.86
5	3.85	125.263	14	1954.89	0.470	2.69	898.55	107.75
6	6.36	88.045	20	785.71	0.561	1.26	702.26	107.84
7	3.54	161.729	16	1101.50	0.218	1.46	933.83	132.07
8	4.25	171.879	14	1977.44	0.239	0.81	261.15	111.55
9	4.87	207.969	18	1992.48	0.495	1.32	761.80	112.69
10	4.88	151.579	18	691.73	0.263	0.81	155.29	120.47
11	3.70	138.796	14	1251.88	0.361	1.45	528.02	113.38
12	4.77	154.962	14	1323.31	0.400	1.30	199.40	98.25
13	5.35	186.917	20	1740.6	0.571	1.49	294.24	109.91
14	4.41	219.624	14	1033.83	0.555	1.71	234.69	101.80
15	3.46	184.285	12	1462.41	0.478	1.24	417.74	99.74
16	5.16	173.759	18	1319.55	0.567	2.40	133.23	114.03
17	6.51	170.000	20	635.34	0.510	0.86	477.29	96.74
18	4.11	153.458	12	943.61	0.136	0.87	340.55	104.83
19	6.62	192.932	20	563.91	0.377	1.27	157.49	108.25
20	5.82	203.082	20	917.29	0.421	2.42	907.37	108.25

**Table 4 materials-14-04790-t004:** Mechanical properties of five kinds of sheets.

Material Parameter	HC220YD	304	5182	DP980	H62
*t*/mm	0.66	0.98	1.2	1.52	1.98
*E*/MPa	167,187	193,358	70,724	218,346	114,649
$\sigma_{s}$/MPa	195	273	123	717	215
*K*/MPa	602	1974	563	1532	778
*n*	0.235	0.590	0.332	0.140	0.345
$r_{0}$	1.50	1.00	0.57	0.83	0.92
$r_{45}$	1.86	1.34	0.64	0.88	1.07
$r_{90}$	2.11	0.87	0.66	0.85	0.89

**Table 5 materials-14-04790-t005:** Forming angles in bending machine experiments.

Material	Number	Punch Stroke	Forming Angle
HC220YD	1	4.22	96.015
	2	3.93	101.380
	3	3.65	106.555
	4	3.38	111.005
304	5	4.07	91.295
	6	3.89	94.155
	7	3.63	99.155
	8	3.35	104.955
5182	9	4.15	91.915
	10	3.87	96.835
	11	3.60	102.095
	12	3.33	107.585
DP980	13	4.08	99.805
	14	3.80	104.885
	15	3.52	109.985
	16	3.26	115.045
H62	17	4.09	91.135
	18	3.82	95.715
	19	3.57	100.575
	20	3.20	107.925

**Table 6 materials-14-04790-t006:** Comparison between testing samples and model predictions.

Number	Finite Element Testing (mm)	Network Prediction (mm)	Stroke Deviation (mm)	Number	Bending Experiment (mm)	Network Prediction (mm)	Stroke Deviation (mm)
1	6.02	6.00	−0.02	21	4.22	4.24	0.02
2	3.97	3.89	−0.08	22	3.93	3.95	0.02
3	4.68	4.59	−0.09	23	3.65	3.67	0.02
4	3.26	3.22	−0.04	24	3.38	3.43	0.05
5	3.85	3.92	0.07	25	4.07	4.17	0.10
6	6.36	6.24	−0.12	26	3.89	4.02	0.13
7	3.54	3.43	−0.11	27	3.63	3.78	0.15
8	4.25	4.23	−0.02	28	3.35	3.51	0.16
9	4.87	4.77	−0.10	29	4.15	4.30	0.15
10	4.88	4.92	0.04	30	3.87	4.01	0.14
11	3.70	3.62	−0.08	31	3.6	3.72	0.12
12	4.77	4.92	0.15	32	3.33	3.44	0.11
13	5.35	5.29	−0.06	33	4.08	4.02	−0.06
14	4.41	4.27	−0.14	34	3.8	3.77	−0.03
15	3.46	3.54	0.08	35	3.52	3.55	0.03
16	5.16	5.10	−0.06	36	3.26	3.36	0.10
17	6.51	6.47	−0.04	37	4.09	4.10	0.01
18	4.11	3.95	−0.16	38	3.82	3.85	0.03
19	6.62	6.66	0.04	39	3.57	3.60	0.03
20	5.82	5.75	−0.07	40	3.2	3.26	0.06

**Table 7 materials-14-04790-t007:** Modeling samples for the correction model.

Number	ΔD/mm	*E/*GPa	$B_{V}/\mathrm{mm}$	*K/*MPa	*n*	*t/*mm	$\sigma_{s}/\mathrm{MPa}$	$\Delta \alpha^{0}$
1	0.1220	75.263	20	695.49	0.369	1.43	510.38	1.257
2	0.2575	116.992	20	1507.52	0.136	2.72	843.41	−2.348
3	−0.2161	77.142	16	1218.05	0.297	1.63	126.62	2.886
4	0.0248	171.503	18	1342.11	0.148	1.95	344.96	−0.269
5	−0.1643	115.112	16	1390.98	0.425	0.73	263.36	2.518
6	−0.1927	97.067	20	921.05	0.542	1.88	772.83	1.621
7	−0.0630	104.210	14	1563.91	0.516	1.85	891.93	1.104
8	−0.0909	166.992	12	1278.20	0.110	1.17	437.59	1.593
9	−0.1181	116.616	16	1078.95	0.212	1.85	534.64	1.350
10	0.0927	153.834	12	1943.61	0.309	1.72	982.36	−1.787
11	0.1320	168.496	18	1887.22	0.182	2.00	618.45	−1.544
12	0.0710	99.323	20	1526.32	0.342	1.44	201.60	−0.938
13	0.2206	122.631	20	1424.81	0.225	2.80	953.68	−2.241
14	−0.1718	143.684	20	1578.95	0.149	2.84	889.72	1.849
15	−0.2088	194.812	14	1913.53	0.248	1.37	631.68	2.934
16	0.1603	111.353	20	1011.28	0.108	2.69	192.78	−1.794
17	−0.0602	196.315	14	1921.05	0.267	1.08	146.47	0.886
18	0.1436	191.052	20	902.26	0.103	1.02	375.84	−1.350
19	0.0297	155.338	12	1116.54	0.491	0.87	847.82	−0.470
…	…	…	…	…	…	…	…	…
1712	0.2450	189.172	18	1443.61	0.128	1.50	532.43	2.854

**Table 8 materials-14-04790-t008:** Testing samples for the correction model.

Number	ΔD/mm	*E/*GPa	$B_{V}/\mathrm{mm}$	*K/*MPa	*n*	*t/*mm	$\sigma_{s}/\mathrm{MPa}$	$\Delta \alpha^{0}$
1	−0.2489	114.736	18	1778.2	0.425	1.75	589.77	2.972
2	−0.0265	123.007	12	1330.83	0.432	1.15	554.49	0.518
3	0.1956	107.969	16	1872.18	0.524	1.01	298.65	−2.487
4	−0.0270	103.834	12	511.28	0.504	0.78	137.64	0.588
5	−0.1048	125.263	14	1954.89	0.470	2.69	898.55	1.666
6	−0.0826	88.045	20	785.71	0.561	1.26	702.26	0.851
7	−0.1668	161.729	16	1101.50	0.218	1.46	933.83	2.698
8	0.1717	171.879	14	1977.44	0.239	0.81	261.15	−2.962
9	−0.1426	207.969	18	1992.48	0.495	1.32	761.80	2.036
10	0.1435	151.579	18	691.73	0.263	0.81	155.29	−1.745
11	0.1432	138.796	14	1251.88	0.361	1.45	528.02	−2.425
12	0.1994	154.962	14	1323.31	0.400	1.30	199.40	−2.768
13	0.0924	186.917	20	1740.6	0.571	1.49	294.24	−1.052
14	0.0955	219.624	14	1033.83	0.555	1.71	234.69	−1.499
15	0.1055	184.285	12	1462.41	0.478	1.24	417.74	−2.172
16	−0.1550	173.759	18	1319.55	0.567	2.40	133.23	1.804
17	−0.2638	170.000	20	635.34	0.510	0.86	477.29	2.716
18	−0.1605	153.458	12	943.61	0.136	0.87	340.55	2.744
19	0.2362	192.932	20	563.91	0.377	1.27	157.49	−2.480
20	0.1653	203.082	20	917.29	0.421	2.42	907.37	−1.894

**Table 9 materials-14-04790-t009:** Parameters of the GA.

Population	Iteration	Selection Operator	Crossover Operator	Mutation Operator
100	200	0.08	0.8	0.03

**Table 10 materials-14-04790-t010:** Bending tests of the correction model.

Mat	Num	Target	$D_{1}$	$\alpha^{0}{}_{1}$	$\Delta \alpha^{0}{}_{1}$	$\Delta D_{1}$	$\alpha^{0}{}_{2}$	$\Delta \alpha^{0}{}_{2}$	$\Delta D_{2}$	$\alpha^{0}{}_{3}$	$\Delta \alpha^{0}{}_{3}$
HC220YD	1	96	4.460	97.745	−1.745	4.578	96.190	−0.190			
	2	100	4.252	101.520	−1.520	4.364	100.150	−0.150			
	3	105	3.973	106.390	−1.390	4.084	105.340	−0.340			
304	4	98	4.050	100.780	−2.780	4.208	99.080	−1.080	4.276	98.195	−0.195
	5	110	3.523	111.940	−1.940	3.644	100.350	−0.135			
	6	120	3.314	119.960	0.040						
5182	7	98	4.168	100.510	−2.510	4.327	98.885	−0.885	4.415	98.155	-0.155
	8	104	3.835	106.470	−2.470	3.990	104.255	−0.255			
	9	110	3.539	112.710	−2.710	3.698	109.640	−0.460			

## Data Availability

No new data were created or analyzed in this study. Data sharing is not applicable to this article.

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
