# Peer review of "A Versatile Punch Stroke Correction Model for Trial V-Bending of Sheet Metals Based on Data-Driven Method"

_materials, 2021, doi:10.3390/ma14174790_

Round 1
Reviewer 1 Report
In this work Y Yu et al. implemented a NN algorithms in the punch stroke correction in metallic structures. The use of NN in this task is not new (see for example https://www.mdpi.com/1996-1944/13/14/3129), however the approach used here is a little different and presents more detailed descriptions. I also think that their approach can be very useful for people starting with NN models as well as, to researchers working in this field.
The article can be accepted after the authors answer the following questions:
As you known the amount of data to test a NN is crucial and in general a very large dataset is desired (>> thousands of samples), however in this work the authors use a very limited number of training set (1712). Here rise my questions:
- How do the authors can ensure that the NN is not overfitting the samples?
- Do they perform a training and validation curve loss as function of number of epochs? please include and explain the curves
- How strong is their NN with respect to large random noise?
I think in order to achieve a larger audience, it is important to include and example of input and out files as a supporting information (e.g, txt files).
I also missed the information of the hardware and software and a summary of the used model, e.g.:
- input shape,
- first layers type and number of variables,
- Output shape
- Second layer type and numbers of variables
- ...
- ...
- Output layer files
Total number of parameters, number of trainable parameters, etc..
Reviewer 2 Report
In this manuscript, the authors have presented how to predict punch stroke correction trials using data-driven ANN method with evolutionary algorithms. The introduction of the manuscript was short but decently written. The authors then presented research methods, where genetic algorithms have been taken into account immediately. However, in the later parts of the study, I did not find any information about used parameters etc. which are most likely to be used inside of the procedure. Back propagation and dimensional analysis was described decently. Best part of this study is tables that compares FEM analysis results with results from machine learning methods. However, because GA-BPNN is so on focus of this study, I would not accept this manuscript before much more descriptive information related to method in use.
Round 2
Reviewer 1 Report
The article can be accepted as it is
Reviewer 2 Report
I recommend to publish this manuscript in the journal of Materials.